# A Unified Understanding and Generation Framework for Ego-Centric Tracing in Dynamic World

## Abstract

Ego-centric tracing with sparse yet informative cues is a fundamental capability of embodied agents operating in complex and dynamic environments. However, existing approaches typically address cue understanding and cue generation in isolation, which limits their synergy and significantly constrains agents' ability to perceive and act effectively. To overcome this limitation, we propose a **Uni**fied **U**nderstanding–**G**eneration framework (**Uni-UG**) that tightly integrates a multi-granularity disentangled representation learning module for understanding with a controllable clue generation module. Specifically, a shared encoder first extracts features from multimodal inputs and interactive feedback, while a temporal attention mechanism dynamically adapts the representation to the evolving environment. The understanding module then disentangles these features into multi-granular sub-representations, capturing rich categorical and fine-grained attribute-level information of potential clues. Conditioned on these outputs and specified control signals, the generation module produces supplementary clue information. A joint loss function is employed to simultaneously optimize understanding accuracy and generation quality, thereby enforcing semantic consistency between the two: the understanding module guides clue generation through extracted categories, while the generated clues in turn iteratively refine the overall understanding process. Extensive experiments conducted across multiple challenging datasets validate the effectiveness and generalizability of Uni-UG framework.

## 1 Introduction

Ego-centric tracking by embodied agents, particularly Unmanned Aerial Vehicles (UAVs), has become a key technological requirement in various dynamic scenarios, such as wildlife monitoring, environmental surveillance, and urban security. In these scenarios, agents often adopt a human-like first-person perspective to perceive their surroundings, interpret environmental cues, and make decisions to execute their tasks. These ego-centric tracking tasks are essential for supporting long-horizon perception and reasoning in dynamic scenarios.

In recent years, ego-centric target tracking has garnered increasing attention. To advance the field, researchers have developed simulation platforms and released benchmark navigation tasks. Given that most vision-and-language navigation (VLN) tasks focus on ground-based robots, the aerial domain has remained relatively underexplored. To address this gap, the AerialVLN (Liu et al., 2023) is introduced, specifically designed for UAVs. CityNav (Lee et al., 2024) guides aerial agents through real urban environments using visual and linguistic cues. Similarly, EmbodiedCity (Gao et al., 2024) focuses on reasoning and tracking within large-scale city settings. NavAgent (Liu et al., 2024c) leverages vision-language models to enable autonomous UAV navigation by integrating multi-scale environmental information. AeroVerse (Yao et al., 2024) aims to enhance the perception, cognition, and action capabilities of aerial and space-based systems, fostering ego-centric interactions among agents, humans, and the environment. Collectively, these efforts share the goal of empowering agents to locate and tracing targets in dynamic environments using multimodal clues.

Despite recent progress, significant challenges remain in applying ego-centric methods to real-world dynamic traceability tasks. **First**, task-relevant clues in dynamic environments are often sparse,

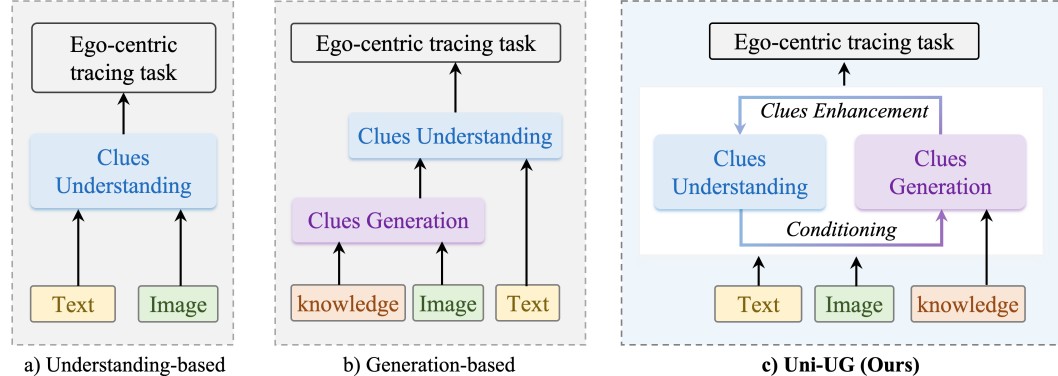

Figure 1: Distinction between existing approaches and the proposed Uni-UG framework. **a)** illustrates methods that rely solely on video understanding without generative support. **b)** shows approaches that introduce predefined generative conditions to produce auxiliary cues, thereby aiding the understanding process. **c)** illustrates our Uni-UG framework, which uses understanding-derived cues as controllable generative conditions to refine existing cues, establishing bidirectional synergy between understanding and generation.

and unevenly distributed, making it difficult for agents to extract critical information and formulate effective decision from limited observations. For instance, in post-disaster search-and-rescue, a UAV may receive only a voice command or a blurry visual cue. Extracting multi-granular semantic information from such weak signals and performing reliable reasoning continues to be a major challenge. **Second**, there is a severe shortage of real-world training data, which limits the development of robust decision-making models using conventional supervised learning. Automatically generating supplementary training data from both historical and current observations is therefore essential for improving model performance. **Third**, most existing methods focus solely on either clue understanding or clue generation, without establishing a synergistic connection between the two. This disconnect often leading to inefficient information use and suboptimal or failed navigation in dynamic environments.

To address these challenges, we propose a **Uni**fied **U**nderstanding and **G**eneration (**Uni-UG**) framework, which comprises a **D**ecoupled **U**nderstanding **M**odule (**DUM**) and a **C**ontrollable **G**eneration **M**odule (**CGM**), as illustrated in Fig 2. Uni-UG tightly integrates the understanding and generation processes through a shared encoder and a joint training strategy. Specifically, the shared encoder extracts key features from multimodal sensory inputs and interaction feedback, with temporal attention mechanisms modeling state evolution in dynamic environments. The DUM decouples these features to identify the categories and attributes of task-relevant clues, while the CGM produces supplementary clues based on the decoupled representations and task-specific conditions. Through iterative interaction, the generated clues are fed back into the DUM, establishing a mutually reinforcing loop that enhances both components over time. The entire framework is optimized using a joint loss function that simultaneously improves understanding accuracy and generation quality. Additionally, we introduce a Direct Preference Optimization (DPO) strategy to guide the clue output distribution toward higher-quality results, and incorporate a temporal consistency constraint to ensure smooth progression of feature representations across time. Extensive experiments across multiple datasets demonstrate the effectiveness of the proposed Uni-UG framework.

Our contributions can be summarized as follows:

- A unified understanding and generation framework for ego-centric tracing is presented, effectively mitigating the challenges posed by sparse, ambiguous, and often unreliable informative cues, while enabling robust perception and reasoning in complex dynamic environments.

- The proposed Uni-UG framework consists of a multi-granularity decoupled understanding module for clue perception and a controllable generation module, with both integrated through parameter sharing and optimized jointly using direct preference optimization.

- Extensive experiments conducted on multiple public benchmarks consistently demonstrate the superior effectiveness, robustness, and generalizability of the Uni-UG framework.

## 2 RELATED WORKS

### 2.1 EMBODIED SCENE UNDERSTANDING

Embodied scene understanding studies how agents interact with their environment to gather multi-modal sensory data for semantic interpretation and decision-making in dynamic settings. Recent progress has moved research from simulation-based environments to scanned real-world indoor scenes, incorporating multiple modalities to reduce the sim-to-real gap (Wang et al., 2024b; Hong et al., 2024). However, these efforts mostly focus on indoor spaces and face challenges when applied to dynamic outdoor environments. Some recent works (Zhu et al., 2024; Gao et al., 2024) address outdoor scenarios but mainly concentrate on urban areas with well-defined physical rules.

Large language models have shown promise in interpreting visual inputs through text, yet they require large-scale data, which is limited in embodied contexts. To overcome this, generative methods have been applied to create synthetic scenes (Yang et al., 2024a), but these focus on structured indoor environments and often overlook challenges in unstructured, dynamic outdoor settings, such as the absence of well-defined physical rules. Another line of work uses generated textual descriptions to represent scenes and produce prompt-based instructions (Yang et al., 2024b; Kong et al., 2024; Lai et al., 2024), but these often ignore the physical dynamics critical for real-world tasks applicability.

In contrast, our work addresses dynamic environments with an emphasis on specific target tracing. Operating under sparse clue constraints, we combine disentangled representation learning with controllable generation techniques to improve perception of scene-level clues. This enables effective and efficient target search in challenging dynamic world.

### 2.2 EGO-CENTRIC TRACING IN THE WILD

Recently, UAV tracing has attracted significant research attention. Anderson et al.(Anderson et al., 2018) first introduced navigation via instruction following in discrete indoor environments. Subsequently, the R2R dataset was extended(Jain et al., 2019) by concatenating adjacent trajectories to generate longer and more complex instructions. In real-world scenarios, multi-turn natural language communication is both common and essential for effective navigation. To simulate this, Thomason et al.(Thomason et al., 2020) collected the CVDN dataset featuring human-to-human dialogue in home environments, tasking agents with navigation based on rich dialogue history. Qi et al.(Qi et al., 2020) introduced remote object grounding and navigation tasks through multiple related datasets to further advance this area.

For UAV navigation in complex urban settings, the AerialVLN dataset (Liu et al., 2023) was proposed, containing 100 diverse flight scenarios across 10 major cities with high-resolution panoramic UAV images. The OpenUAV platform (Wang et al., 2024c) facilitates realistic and scalable UAV vision-and-language navigation tasks. CityNav (Lee et al., 2024) provides a city-scale aerial VLN dataset that demands advanced planning, high-level spatial reasoning, and robust decision-making. AeroVerse (Yao et al., 2024) addresses a critical research gap in UAV embodied world modeling, significantly enhancing end-to-end autonomous perception, cognition, and action capabilities.

In this work, we focus on first-person UAV trajectory tracing and demonstrate promising results across multiple datasets using the proposed Uni-UG framework.

## 3 UNI-UG FRAMEWORK

Autonomous tracing in dynamic environments reflects an agent's perceptual and decision-making capabilities. However, sparse informative cues in such settings make it challenging to extract key features from limited sensory input, hindering tracking performance. Most existing methods treat cue understanding and generation independently, lacking integration. This limits their ability to fully model and respond to dynamic scenes.

To overcome this, we propose the **Uni**fied **U**nderstanding and **G**eneration (**Uni-UG**) framework, which jointly optimizes cue understanding and generation in a closed-loop system. This enables both modules to reinforce each other, improving scene modeling and decision-making. Specifically, the **D**ecoupled **U**nderstanding **M**odule (**DUM**) uses a shared encoder and temporal attention to ex-

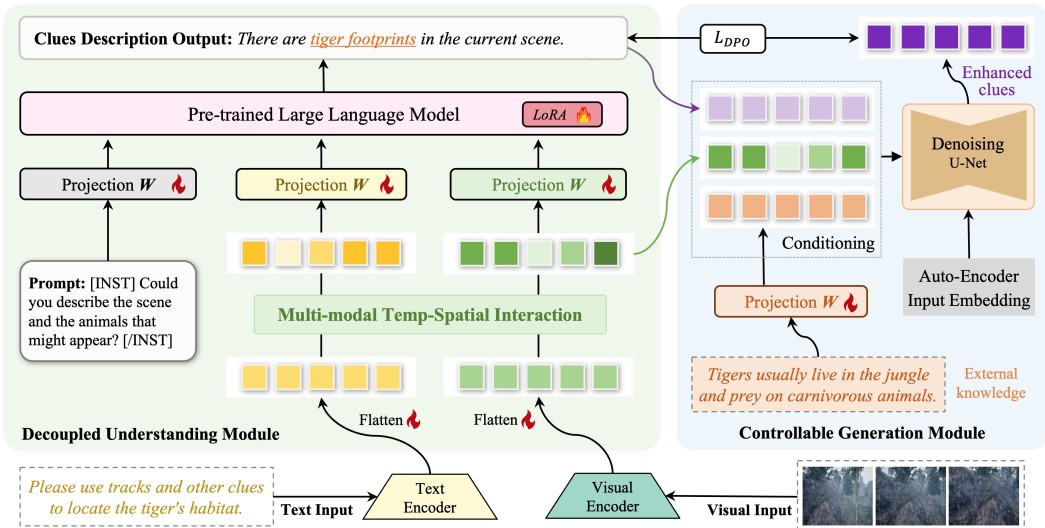

Figure 2: The proposed Uni-UG framework. Multimodal inputs are first encoded for feature extraction and processed by the Cue Understanding Module for initial semantic modeling. The Decoupled Understanding Module identifies critical sparse cues, which condition the Controllable Generation Module to produce supplementary clues, enhancing overall scene understanding. The two modules are jointly optimized via direct preference optimization, enabling mutual reinforcement between understanding and generation, significantly improving environmental modeling and decision-making.

tract and separate multi-granular features from multimodal inputs. These are used to infer latent cue categories and attributes. The **C**ontrollable **G**eneration **M**odule (**CGM**) leverages the outputs from DUM along with control conditions to generate supplementary cues, enriching the agent's scene representation. Moreover, the two modules are jointly optimized using Direct Preference Optimization, reinforcing synergy between understanding and generation, and improving the agent's object tracing performance in dynamic environments.

## 3.1 MULTIMODAL TOKENIZATION

Given the task description $I_{task}$ and the instruction $I_{instr}$, we first tokenize both using a pre-trained language tokenizer to obtain the corresponding textual tokens. For multimodal inputs, we adopt the EVA-CLIP and Q-former to extract visual features. Each image is transformed into a set of tokens, consisting of one context token that captures global information and 16 content tokens that represent local details via grid pooling. For other modalities inputs, we apply the same processing pipeline as used for RGB images to obtain their token representations. Finally, we concatenate all token types, including image tokens $T_{img}$, task description tokens $T_{task}$, and instruction tokens $T_{instr}$, to construct the final multimodal input token sequence, denoted as $T_{input} = <T_{img}, T_{task}, T_{instr}>$.

## 3.2 DECOUPLED UNDERSTANDING MODULE

In dynamic world, reward cues are often sparse and unevenly distributed. Traditional works struggle to capture their diversity and fail to meet the demands of complex tasks. This project proposes a continual decoupled understanding method based on multi-granularity representations, which incrementally separates features from coarse-grained (e.g., cue existence) to fine-grained (e.g., cue category) levels. The method progressively enhances the independence and task relevance of each representation, thereby improving the agent's ability to interpret dynamic and sparse reward signals.

Specifically, the input multimodal data is first processed through an encoder to obtain initial features $f_0$, representing the state at the current time step. For features at different levels of granularity that span coarse to fine semantic details, a multi-layer decoupling strategy is adopted. Features are decomposed into multiple sub-representations, with each layer focusing on a specific granularity

level, isolating entangled information, and applying a linear transformation as follows:

$$f_l = W_l f_{l-1} + b_l, \tag{1}$$

where $W_l$ is the transformation matrix, $b_l$ is the bias term, and $f_l$ represents the features at granularity level $l$.To adapt to environmental dynamics, continual optimization is introduced. Sub-representations are updated dynamically via orthogonality constraints that prevent feature overlap and task-oriented loss functions:

$$\mathcal{L}_{\text{layer}} = \sum_l \lambda_l \cdot \mathcal{L}_{\text{task}}(h_l(f_l), y_l) + \mathcal{L}_{\text{orth}}(f_l), \tag{2}$$

where $h_l$ and $y_l$ is the task head and ground-truth label at level $l$, respectively, $\lambda_l$ is the weight coefficient for each layer, $\mathcal{L}_{\text{task}}$ denotes a task loss (e.g., cross-entropy or MSE), $\mathcal{L}_{\text{orth}}$ is an orthogonality constraint promoting disentanglement. To maintain temporal consistency under environmental changes, we introduce a time-consistency loss:

$$\mathcal{L}_{\text{temporal}} = \|f_t - f_{t-1}\|^2. \tag{3}$$

The overall loss for DUM is formulated as:

$$\mathcal{L}_{\text{DUM\_total}} = \mathcal{L}_{\text{decouple}} + \alpha \cdot \mathcal{L}_{\text{temporal}}. \tag{4}$$

where $\alpha$ is the temporal weighting factor. By using multi-granularity representations and continual decoupling, this method effectively isolates reward cues at different levels, addressing the challenge of sparsity and providing a solid foundation for downstream decision-making and causal tracing.

### 3.3 CONTROLLABLE GENERATION MODULE

Sparse reward cues in dynamic environments limit an agent's ability to make effective decisions and trace the source of rewards. To address this issue, we design a controllable generation module (CGM) conditioned on the outputs of the DUM. The CGM module leverages interaction feedback from the environment to generate fine-grained reward cues, thereby enhancing the agent's understanding of dynamic surroundings. First, given multimodal input data $x$ and interaction feedback $g$, we extract their featur representations via:

$$z = \text{Enc}_x(x), \quad g = \text{Enc}_g(g), \tag{5}$$

where $Enc(\cdot)$ is feature encoder. Then, using a weighted attention mechanism, the two input are fused to obtain a joint conditional representation:

$$c = \sum_i \alpha_i \cdot \text{Enc}_{\text{cond}}(z_i, g_i), \tag{6}$$

where $\alpha_i$ is attention weights, $Enc_{\text{cond}}$ is the conditional encoder. A diffusion model with a U-Net architecture is used as the generator to iteratively refine and produce new reward cues based on the fused multi-granular features:

$$\hat{y} \sim p_\theta(y \mid c), \tag{7}$$

where $p_\theta$ is the conditional generation distribution, and $\hat{y}$ is the generated cue constrained to match both the environment and control conditions. The loss for controllable generation is defined as:

$$\mathcal{L}_{\text{gen}} = \beta_1 \cdot \mathcal{L}_{\text{MSE}}(\hat{y}, y) + \beta_2 \cdot \mathcal{L}_{\text{cond}}(\hat{y}, c), \tag{8}$$

where $\mathcal{L}_{\text{MSE}}$ is the pixel-level mean squared error, $\mathcal{L}_{\text{cond}}$ ensures consistency with control conditions (e.g., cross-entropy), $\beta_1, \beta_2$ are balancing coefficients.

In summary, by integrating environmental interaction feedback with controllable generation, this produces richer informative reward cues, guiding agents to better understand their relationship with the environment and improving decision-making performance.

## 3.4 Unified Understanding-Generation

In the preceding modules, the DUM first employs disentangled representation learning to extract multi-granular cue features. These features are then linearly transformed and used as conditions for the CGM to generate reward-related cues via a diffusion model. However, a key challenge remains: how to jointly optimize DUM and CGM to enable mutual enhancement between understanding and generation. To address this, we adopt *Direct Preference Optimization (DPO)* to jointly train both modules, encouraging the agent to effectively model sparse cues in dynamic environments and thereby improve overall performance.

Following the previous modules, we have already obtained the shared input feature representation $f$, multi-granularity disentangled representations $\{f_1, f_2, \ldots, f_L\}$, and the output from the DUM:

$$o = \text{Decoder}(f_L), \quad o = \{\hat{c}, \hat{a}\}, \tag{9}$$

where $\hat{c}$ and $\hat{a}$ denote the predicted class and attributes at the finest granularity level, respectively, and the decoder parameters remain fully trainable. These outputs from the DUM serve as informative conditional signals to guide the subsequent generation process:

$$\hat{y} \sim p_\theta(y \mid f, \hat{c}, \hat{a}), \tag{10}$$

where the generator takes the shared features $f$ and the understanding output $\hat{c}, \hat{a}$ as input to generate new reward cues, where $p_\theta$ is the conditional generation distribution. To ensure consistency between understanding and generation while maintaining the independence of disentangled representations, we formulate a unified joint optimization objective as follows:

$$\mathcal{L}_{\text{joint}} = \mathcal{L}_{\text{gen}}(y, \hat{y}) + \mathcal{L}_{\text{DUM\_total}} + \mathcal{L}_{\text{orth}}, \tag{11}$$

where $\mathcal{L}_{\text{gen}}$ is the generation loss (e.g., MSE or diffusion-specific loss), $\mathcal{L}_{\text{DUM\_total}}$ is the understanding loss (e.g., classification and attribute prediction), $\mathcal{L}_{\text{orth}}$ is an orthogonality loss to maintain the disentanglement of features. To optimize the joint training of the DUM and CGM modules, DPO is introduced as strategy:

$$\mathcal{L}_{\text{DPO}} = \log \frac{\sigma((s^+ - s^-)/\tau)}{1 + \sigma((s^+ - s^-)/\tau)}, \tag{12}$$

where $s^+$, $s^-$ represent the scores for preferred and less preferred cues, $\tau$ is the temperature parameter, $\sigma(\cdot)$ is the Sigmoid function. This encourages the model to assign higher generation probabilities to superior reward cues over inferior ones. The DPO is computed relative to a reference model (e.g., a pre-trained baseline). The final loss for the unified understanding-generation framework is defined as follows:

$$\mathcal{L}_{\text{total}} = \mathcal{L}_{\text{joint}} + \gamma \cdot \mathcal{L}_{\text{DPO}}, \tag{13}$$

where $\gamma$ is the weighting coefficient for preference optimization. Moreover, the generated reward cues $\hat{y}$ can be fed back as additional input for the next round of DUM, forming a closed-loop learning mechanism that enhances adaptive performance over iterations by refining feature representations and aligning them more closely with task objectives.

In conclusion, the proposed Uni-UG framework establishes a unified and adaptive mechanism that not only generates fine-grained reward cues from interpreted information but also continuously refines its understanding by leveraging these generated cues in return. This bi-directional synergy creates a closed-loop semantic modeling process, enabling more robust, explainable, and accurate tracing, perceptual reasoning, and decision-making in dynamic real-world environments characterized by sparse and ambiguous reward cues.

## 4 Experiments

### 4.1 Dataset

**AirVLN** (Liu et al., 2023) is a dataset for aerial VLN, where agents follow language instructions to navigate from a start point to a goal. It spans 25 city-scale environments, including downtowns,

Table 1: Performance of the proposed Uni-UG framework on the UrbanVideo-Bench dataset.

| Method | Avg. | Recall | | | | | Perception | | | | | Reasoning | | | Navigation | | |
|---|---|---|---|---|---|---|---|---|---|---|---|---|---|---|---|---|---|
| | | TCap | SeR | OR | ScR | SP/EP | Prox | Dur | LandP | GlD | CMap | Cau | Cnt | Assoc | PEval | HighP | ActGen |
| Gemini-1.5-Flash (Team et al., 2024) | 40.5 | 39.7 | 51.8 | 61.7 | 79.3 | 61.3 | 47.1 | 59.8 | 37.8 | 28.7 | 47.9 | 60.0 | 42.4 | 20.0 | 43.3 | 32.6 | 34.4 |
| Gemini-1.5-Pro (Team et al., 2024) | 42.5 | 58.6 | 61.6 | 65.0 | 72.1 | 66.2 | 66.4 | 63.6 | 37.4 | 33.8 | 46.0 | 63.6 | 46.2 | 23.0 | 38.8 | 43.8 | 31.9 |
| Gemini-2.0-Flash (Google, 2025) | 38.3 | 47.9 | 58.9 | 63.3 | 75.7 | 57.0 | 66.4 | 47.7 | 27.9 | 27.8 | 45.3 | 62.7 | 24.2 | 17.8 | 39.2 | 48.4 | 30.5 |
| GPT-4o-mini (OpenAI, 2025) | 36.5 | 33.0 | 53.6 | 48.3 | 59.5 | 56.3 | 69.7 | 51.5 | 33.3 | 31.3 | 42.4 | 65.5 | 47.7 | 22.9 | 30.8 | 57.5 | 25.4 |
| GPT-4o (OpenAI, 2025) | 43.6 | 47.6 | 58.9 | 65.0 | 67.6 | 61.3 | 63.0 | 47.7 | 36.8 | 42.4 | 52.8 | 66.4 | 44.7 | 45.8 | 34.2 | 67.8 | 33.8 |
| Qwen-VL-Max-latest (Cloud, 2025) | 45.5 | 44.9 | 70.5 | 64.2 | 75.7 | 73.9 | 78.2 | 43.9 | 44.8 | 44.7 | 61.1 | 77.3 | 49.2 | 23.9 | 38.8 | 70.0 | 29.6 |
| LLaVA-NeXT-Video-7B (Liu et al., 2024a) | 38.6 | 55.7 | 39.3 | 43.3 | 61.3 | 40.8 | 58.8 | 52.3 | 49.5 | 16.7 | 26.8 | 44.5 | 20.5 | 58.7 | 36.6 | 52.3 | 19.2 |
| Phi-3.5-vision-instruct (Abdin et al., 2024) | 38.7 | 67.0 | 57.1 | 57.5 | 64.9 | 45.1 | 48.7 | 45.5 | 49.2 | 17.0 | 52.1 | 51.8 | 34.8 | 13.9 | 33.2 | 59.7 | 15.6 |
| Kangaroo (Liu et al., 2024b) | 39.2 | 27.0 | 66.1 | 60.8 | 69.4 | 53.5 | 75.6 | 57.6 | 35.5 | 37.2 | 60.0 | 64.5 | 42.4 | 19.1 | 32.5 | 41.9 | 32.4 |
| InternVL2-2B (Chen et al., 2024) | 27.6 | 19.2 | 29.5 | 37.5 | 55.9 | 22.5 | 57.1 | 37.9 | 19.3 | 24.6 | 39.2 | 33.6 | 45.5 | 33.5 | 29.2 | 37.6 | 20.9 |
| InternVL2-4B (Chen et al., 2024) | 28.1 | 19.2 | 37.5 | 33.3 | 62.2 | 24.6 | 66.4 | 42.4 | 23.2 | 26.5 | 32.8 | 36.4 | 35.6 | 24.8 | 29.5 | 32.2 | 22.1 |
| InternVL2-8B (Chen et al., 2024) | 28.1 | 23.4 | 23.2 | 35.0 | 52.3 | 22.5 | 58.0 | 44.7 | 23.1 | 27.4 | 28.3 | 33.6 | 45.5 | 27.0 | 31.5 | 35.7 | 21.4 |
| Qwen2-VL-2B-Instruct (Wang et al., 2024a) | 31.9 | 29.9 | 54.5 | 30.8 | 57.7 | 24.6 | 69.7 | 47.7 | 22.0 | 22.1 | 64.2 | 46.4 | 35.6 | 13.5 | 28.8 | 44.2 | 27.3 |
| Qwen2-VL-7B-Instruct (Wang et al., 2024a) | 36.2 | 36.5 | 50.9 | 47.5 | 65.8 | 47.2 | 52.1 | 48.5 | 25.1 | 28.4 | 55.8 | 55.5 | 29.5 | 11.7 | 33.9 | 59.3 | 32.7 |
| Uni-UG (Ours) | 39.6 | 37.1 | 52.0 | 48.3 | 66.7 | 45.8 | 60.4 | 51.5 | 24.8 | 27.4 | 60.1 | 58.7 | 32.4 | 13.8 | 35.2 | 63.6 | 33.4 |

*Note: Trajectory Captioning (TCap), Sequence Recall (SeR), Object Recall (OR), Scene Recall (ScR), Start/End Position (SP/EP), Proximity (Prox), Duration (Dur), Landmark Position (LandP), Goal Detection (GD), Cognitive Map (CMap), Causal (Cau), Counterfactual (Cnt), Association (Assoc), Progress Evaluation (PEval), High-level Planning (HighP), Action Generation (ActGen).

parks, and villages, and features over 870 distinct objects. A total of 8,446 UAV trajectories are collected, each paired with three natural language instructions. In total, the dataset provides 25,338 instructions, averaging 83 words in length and using a vocabulary of 4,470 unique words.

**CityNav** (Lee et al., 2024) offers 32,637 human-demonstrated trajectories linked to 5,850 real-world objects like buildings and vehicles. Collected via a web-based 3D simulator integrated with MTurk, it supports large-scale aerial VLN research across varied urban and suburban environments and is split into train, validation seen/unseen, and test unseen sets.

**UrbanVideo-Bench** (Zhao et al., 2025) contains 1,547 real-world drone videos (1280×720), ranging from 10 seconds to 10 minutes across diverse urban and natural settings. The dataset captures complex 3D flight patterns and provides over 5,200 multiple-choice questions, covering tasks from low-level perception to high-level reasoning and navigation.

## 4.2 EVALUATION METRICS.

Drawing on commonly used evaluation metrics in the VLN domain (Liu et al., 2023; Lee et al., 2024; Zhao et al., 2025; Anderson et al., 2018), this paper adopts several sub-metrics, including Success Rate (SR), Oracle Success Rate (OSR), Success weighted by Path Length (SPL), and Navigation Error (NE). Specifically: *SR*: Measures the proportion of tasks in which the UAV successfully reaches the target location within a predefined tolerance range. *OSR*: Measures whether the UAV reaches any point along the optimal trajectory, even if it does not exactly reach the final destination, accounting for partial success cases. *SPL*: Evaluates both the success of task completion and the efficiency of the path taken, encouraging shorter and more optimal navigation paths relative to ground truth. *NE*: Represents the average Euclidean distance between the UAV's final position and the target location, quantifying precision errors accurately.

## 4.3 IMPLEMENTATION DETAILS

In all experiments, we adopted encoder configurations consistent with those used in the respective baseline models for each task-specific dataset. For general models such as Seq2Seq (Anderson et al., 2018) and CMA (Vaswani et al., 2017), training was performed using the Adam optimizer for 5 epochs, with a learning rate of $1.5 \times 10^{-3}$ and a batch size of 12. For the MGP model (Lee

Table 2: Performance of the proposed Uni-UG framework on the AerialVLN dataset.

| Mehtod | Validation Seen | | | | Validation Unseen | | | | Test Unseen | | | |
|---|---|---|---|---|---|---|---|---|---|---|---|---|
| | NE↓ | SR↑ | OSR↑ | SDTW↑ | NE↓ | SR↑ | OSR↑ | SDTW↑ | NE↓ | SR↑ | OSR↑ | SDTW↑ |
| LingUNet (Misra et al., 2018) | 383.8 | 0.6 | 6.9 | 0.2 | 368.4 | 0.4 | 3.6 | 0.9 | 399.8 | 0.1 | 3.1 | 0.1 |
| Seq2Seq (Anderson et al., 2018) | 146.0 | 4.8 | 19.8 | 1.6 | 218.9 | 2.3 | 11.7 | 0.7 | 214.6 | 2.2 | 9.4 | 0.7 |
| CMA (Vaswani et al., 2017) | 121.0 | 3.0 | 23.2 | 0.6 | 172.1 | 3.2 | 16.0 | 1.1 | 178.5 | 3.9 | 13.1 | 1.4 |
| Seq2Seq-DA (Anderson et al., 2018) | 85.5 | 9.9 | 24.1 | 4.5 | 143.5 | 4.0 | 10.9 | 0.7 | 140.2 | 3.5 | 9.5 | 0.6 |
| CMA-DA (Vaswani et al., 2017) | 92.2 | 9.9 | 26.5 | 3.7 | 122.7 | 4.5 | 13.9 | 1.0 | 125.4 | 4.3 | 14.8 | 1.2 |
| LAG (Liu et al., 2023) | 90.2 | 7.2 | 15.7 | 2.4 | 127.9 | 5.1 | 10.5 | 1.4 | 128.3 | 4.5 | 11.6 | 1.3 |
| **Uni-UG (Ours)** | 88.6 | 8.0 | 22.4 | 3.9 | 122.5 | 5.2 | 14.1 | 1.3 | 123.6 | 4.3 | 15.0 | 1.4 |

Table 3: Performance of the proposed Uni-UG framework on the CityNav dataset. Learning-based models are evaluated with shortest path (SP) or human demonstrations (HD) trajectories.

| Mehtod | Validation Seen | | | | Validation Unseen | | | | Test Unseen | | | |
|---|---|---|---|---|---|---|---|---|---|---|---|---|
| | NE↓ | SR↑ | OSR↑ | SPL↑ | NE↓ | SR↑ | OSR↑ | SPL↑ | NE↓ | SR↑ | OSR↑ | SPL↑ |
| Seq2Seq w/ SP (Anderson et al., 2018) | 148.4 | 4.52 | 10.61 | 4.47 | 201.4 | 1.04 | 8.03 | 1.02 | 174.5 | 1.73 | 8.57 | 1.69 |
| Seq2Seq w/ HD (Anderson et al., 2018) | 257.1 | 1.81 | 7.89 | 1.58 | 317.4 | 0.79 | 8.82 | 0.61 | 245.3 | 1.50 | 8.34 | 1.30 |
| CMA w/ SP (Vaswani et al., 2017) | 151.7 | 3.74 | 10.77 | 3.70 | 205.2 | 1.08 | 7.89 | 1.06 | 179.1 | 1.61 | 10.07 | 1.57 |
| CMA w/ HD (Vaswani et al., 2017) | 240.8 | 0.95 | 9.42 | 0.92 | 268.8 | 0.65 | 7.86 | 0.63 | 252.6 | 0.82 | 9.70 | 0.79 |
| MGP w/ SP (Lee et al., 2024) | 75.0 | 6.53 | 22.26 | 6.27 | 93.4 | 4.32 | 15.00 | 4.24 | 109.0 | 4.73 | 17.47 | 4.62 |
| MGP w/ HD (Lee et al., 2024) | 59.7 | 8.69 | 35.51 | 8.28 | 75.1 | 5.84 | 22.19 | 5.56 | 93.8 | 6.38 | 26.04 | 6.08 |
| **Uni-UG (Ours)** | 55.4 | 9.10 | 36.62 | 8.61 | 71.3 | 6.02 | 23.74 | 5.94 | 89.9 | 6.75 | 29.18 | 6.70 |

et al., 2024), we followed the original paper's setup, employing the AdamW optimizer for 10 epochs with a lower learning rate of $1.0 \times 10^{-3}$ and a smaller batch size of 8. For large-scale pre-trained models involved in our experiments, we used their official APIs for inference or fine-tuning. All experiments were conducted on $6\times$ NVIDIA A100-40G GPUs.

## 4.4 QUANTITATIVE RESULT

To validate the effectiveness of the proposed Uni-UG framework, we conducted comparative experiments against several mainstream methods, including Gemini-1.5 (Team et al., 2024), GPT-4o (OpenAI, 2025), Qwen-VL-Max-latest (Cloud, 2025), InternVL2 (Chen et al., 2024), and others, As shown in Tab 1, the performance of different large models on the UrbanVideo-Bench dataset varies significantly. We selected Qwen2-VL-7B-Instruct (Wang et al., 2024a) as the LLM API for our experiments. The results indicate that Uni-UG achieves substantial improvements across all sub-metrics as well as the overall performance metric. We also observe that the model's performance does not match that of larger models such as GPT-4o (OpenAI, 2025), primarily due to differences in parameter scale. Given the limitations in computational resources, we conducted our experiments using only Qwen2-VL-7B-Instruct (Wang et al., 2024a). Despite this, the results remain compelling, further demonstrating the effectiveness of the proposed Uni-UG framework. Furthermore, the results on the AerialVLN dataset, presented in Tab 2, clearly demonstrate that the proposed framework outperforms other baseline methods on most sub-metrics, showcasing superior navigation and environmental understanding capabilities. In addition, we evaluated our framework on the newly released CityNav dataset. Although the simulation platform for this dataset is not yet publicly available, the original data can be accessed. As shown in Tab 3, Uni-UG also delivers strong performance on CityNav. It is worth noting that our method utilizes human demonstration trajectories in this setting. In summary, the proposed Uni-UG framework achieves consistently strong results across multiple challenging benchmark datasets, providing solid evidence of its effectiveness in dynamic world.

## 4.5 ABLATION STUDIES

In this subsection, we present a comprehensive analysis of the contributions made by each component within the Decoupled Understanding Module (DUM) and the Controllable Generation Module (CGM), using the AerialVLN dataset as a representative case study. As shown in Table 4, removing either the DUM (w/ DUM) or the CGM (w/ CGM) results in a substantial decline in performance across both aggregate and task-specific metrics. This clearly indicates that both modules are integral to the overall effectiveness of the Uni-UG framework. In particular, the absence of the CGM significantly limits the framework's capacity for detailed understanding, as the Uni-UG model must then rely solely on Qwen2-VL-7B-Instruct (Wang et al., 2024a) to produce coarse-grained scene descriptions, without the benefit of context-aware cue generation. This simplification undermines the

Table 4: The impact of different modules within the Uni-UG on performance in the AerialVLN.

| AerialVLN-S | Validation Seen | | | | Validation Unseen | | | | Test Unseen | | | |
|---|---|---|---|---|---|---|---|---|---|---|---|---|
| | NE ↓ | SR ↑ | OSR ↑ | SDTW ↑ | NE ↓ | SR ↑ | OSR ↑ | SDTW ↑ | NE ↓ | SR ↑ | OSR ↑ | SDTW ↑ |
| w/ DUM | 101.4 | 7.4 | 22.0 | 3.7 | 129.3 | 4.8 | 13.5 | 1.3 | 126.6 | 4.1 | 14.0 | 1.1 |
| w/ CGM | 95.1 | 7.7 | 21.2 | 3.5 | 126.4 | 4.9 | 12.3 | 1.2 | 127.4 | 4.0 | 12.9 | 1.2 |
| **Uni-UG (Ours)** | 88.6 | 8.0 | 22.4 | 3.9 | 122.5 | 5.2 | 14.1 | 1.3 | 123.6 | 4.3 | 15.0 | 1.4 |

Figure 3: Visualization results. Subfigures (a) and (b) represent urban and rural scenes in a simulated environment, respectively, while (c) shows a real-world urban scene. The visualizations demonstrate that the Uni-UG framework is capable of capturing critical sparse cues and enabling accurate decision-making and execution by the agent.

agent's ability to model complex, dynamic environments. These ablation results highlight not only the individual importance of the DUM and CGM, but also the synergistic effect achieved through their integration. Collectively, they validate the design choices underlying Uni-UG and demonstrate its robustness in addressing the challenges of UAV Tracing navigation in dynamic real-world scenarios. Overall, the ablation results clearly demonstrate the effectiveness and necessity of the Uni-UG framework and its individual modules in handling dynamic environment tasks.

## 4.6 VISUALIZATION

To further validate the effectiveness of the Uni-UG framework, we conducted comprehensive visualization experiments across a diverse set of scenarios, including simulated urban and rural environments as well as complex real-world urban scenes. As shown in Fig. 3, panels (a) and (b) correspond to the simulated urban and rural settings, respectively, while (c) depicts a real-world urban scene. The results clearly demonstrate that Uni-UG can accurately identify and capture critical sparse cues in dynamic and visually complex environments, thereby substantially enhancing the agent's capability to understand its surroundings and make more informed, context-aware decisions.

## 5 CONCLUSION

In this work, we propose a unified understanding and generation (Uni-UG) framework to address the challenges faced by agents in dynamic world, where sparse cues hinder effective decision-making and execution. Uni-UG comprises a Decoupled Understanding Module (DUM) and a Controllable Generation Module (CGM). The DUM is designed to extract critical cues from dynamic scenes and provide them as control conditions for the CGM, which in turn generates richer cues to further enhance the agent's understanding of the environment. To enable effective synergy between the two modules, we adopt a DPO strategy for joint training, allowing understanding and generation to reinforce each other. Extensive experiments on multiple datasets demonstrate the effectiveness of the proposed Uni-UG framework.

ETHICS STATEMENT

Our work is built upon publicly available datasets that do not contain any sensitive or private information. We have thoroughly reviewed the datasets to ensure that no ethical concerns, such as biased or offensive content, arise from their use. Based on our analysis, we do not anticipate any harmful societal impact or unintended bias resulting from this research. We are committed to ethical standards in research and have ensured that our work aligns with these principles.

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
