# OpenReview forum: "A Unified Understanding and Generation Framework for Ego-Centric Tracing in Dynamic World"
_ICLR.cc/2026/Conference — ICLR 2026 Conference Withdrawn Submission_

### Official Review · Reviewer_J9p6 · 2025-10-21

**Soundness:** 2
**Presentation:** 1
**Contribution:** 2
**Rating:** 2
**Confidence:** 4

**Summary:**

The paper addresses the task of egocentric tracking in UAVs. The work seems to show results across benchmarks that answer questions (based on UAV video trajectories) or instruction following navigation. The author’s core contribution is to a unified paradigm that leverages both understanding-based and generation-based cues to improve model performance. These two modules are also encouraged to interact with each other through a DPO loss, which allows for iterative refinement.

**Strengths:**

- The paper addresses an interesting problem of instruction-guided drone navigation, which has various useful applications.

**Weaknesses:**

- The paper is unfortunately not well written and is difficult to understand. It uses a lot of jargon without clearly defining it anywhere.
   - Egocentric tracing <-> tracking, these terms seem to be used interchangeably, which one is the actual task being addressed? In the final evals, the tasks seem to be VQA-based on drone trajectories or navigation. How are they related to tracking?
   - L052: Ego-centric methods to real-world dynamic traceability tasks -> what egocentric methods are being referred to here, and what exactly are dynamic traceability tasks?
   - Cues <-> clues are used interchangeably. What these clues are defined only sparsely across the text, at some points, they referred to the initial instruction, at some points, they seem to suggest being class attributes, etc.
   - In the intro, L077, it's highlighted that the lack of real-world training data limits the development of decision-making models using supervised learning. Is this data for behavior cloning for policy training? How does it relate to the clue generation depicted in Figure 2? It is also unclear how your method fixes this bottleneck, in terms of the arguments presented in the text or experimental results.
  - In Sec. 3.3, L246, there seems to be something called external feedback, which I was not able to place, where it is exactly defined. Is it the external knowledge in Figure 2, or something else?
  - What is the orthogonality constraint mentioned in L222, and how is it implemented?

- What do the generated clues look like? Are they just latent in a model that feeds back into the understanding LLM? Or is it concrete text instructions, class attributes?
- Lack of analysis: The paper conducts only one analysis, where it removes the understanding and generation module, to isolate its importance. It's a good one, but are the roles of orthogonality, temporal constraints, the gamma parameter remain unexplored. These factors can be useful to provide insights to future users of your work.

**Questions:**

- The architecture and method description are not very clear on the output structure; more clarity on those points would make it easier to follow. For instance, how does this model go into producing actions on the CityNav dataset?
- Writing flow: The second paragraph of the introduction turned into more like a related work. It made the flow a bit awkward. I would suggest moving some of that content to the related work and using that space to set the base of your work better for a first-time reader.

---

### Official Review · Reviewer_Gkvo · 2025-10-26

**Soundness:** 3
**Presentation:** 2
**Contribution:** 2
**Rating:** 2
**Confidence:** 3

**Summary:**

This paper addresses the challenge of ego-centric tracing in complex environments, where embodied agents must effectively interpret and generate sparse yet informative cues. The authors propose a Unified Understanding–Generation framework (Uni-UG) that jointly models cue understanding and generation. The method integrates a multigranularity disentangled representation learning module with a controllable clue generation module, linked through a shared encoder and temporal attention mechanism. A joint optimization objective enforces semantic consistency between understanding and generation, allowing each to improve the other. Experiments on multiple datasets demonstrate that Uni-UG achieves strong performance and generalization, suggesting that tighter integration of understanding and generation can enhance embodied perception and reasoning capabilities.

**Strengths:**

-  This work proposes a novel approach that leverages understanding-derived cues as controllable generative conditions to refine existing cues.
- The paper compares a large number of models and provides multiple metrics to evaluate the performance of Uni-UG.

**Weaknesses:**

- The paper lacks some prominent identifiers in its tables — for instance, the performance scores of methods in Tables 1, 2, and 3 fail to intuitively indicate the top performance within each subcategory.
- Both open-source models compared in the paper, Qwen2-VL and InternVL2, are relatively outdated.
- The paper only analyzes whether the DUM and CGM modules function, without delving into the reasons why these two modules work.

**Questions:**

- The tables in the paper do not clearly highlight top-performing results. Could the authors revise Tables 1–3 to more explicitly indicate the best scores within each subcategory for easier comparison?
- The paper benchmarks Uni-UG against Qwen2-VL and InternVL2, which are relatively outdated models. Why were these particular baselines chosen, and how would the proposed framework compare to more recent vision-language models?
- The ablation studies primarily verify that the DUM and CGM modules contribute to performance improvements, but do not analyze why these components are effective. Could the authors provide deeper analysis or visualization to clarify the underlying mechanisms that make these modules work?

---

### Official Review · Reviewer_KzdA · 2025-11-01

**Soundness:** 3
**Presentation:** 3
**Contribution:** 2
**Rating:** 4
**Confidence:** 4

**Summary:**

This paper introduced Uni-UG to conduct generation and understanding through a shared encoder and a joint training strategy. Those two module get different granularity features for both sides, then the Controllable Generation Module (CGM) generate the clues as in Figure 2. Two modules are optimized with DPO so to do understanding and generation training in dynamics environment. They tested the performance on two navigation benchmarks and one Video drone dataset.

**Strengths:**

1. This paper presented a unified framework for understanding and generation in dynamic environment. By sharing the same encoding, they use DPO to train both parts.

2. They did experiments on several benchmarks in VLN domain, though the exp setting is not clear.

3. They conducted ablation study and visualization, which validated the effectiveness of the components.

**Weaknesses:**

1. In table 1, 2, and 3, the model did not achieve sota performance on many of the metrics, especially on table 1. Also, model used in the table are not sota baselines. For example, gemini 2.0 pro is not used but 2.0 flash. Gemini 2.5 is also available but not tested.

2. This paper claims a unified model but the experiments are only about the understanding part in VLN setting without showing any quantitative results on visual generation.

3. Many details missing. For example, what is the weight gamma used in Eq. 13? Beta in Eq. 8? Why do you need both Beta_1 and Beta_2? The loss ratio could be represented by using only 1 beta. What is the reference model in DPO? No appendix to find out these details.

**Questions:**

1. DPO requires to know which one is the better example. How did you find it? What is the training set?

2. What is the backbone used for Uni-UG? Could you provide more about the experiment setting?

---

### Note · Authors · 2025-11-17

I have read and agree with the venue's withdrawal policy on behalf of myself and my co-authors.